# The impact of new urbanization on water ecological resilience: An empirical study from central China

**Daxue Kan** [1,2]*, **Lianjv Lv**[1,2]

1 School of Economics and Trade, Nanchang Institute of Technology, Nanchang, Jiangxi, China,
2 Nanchang Institute of Technology, Water Economics and Management Research Center, Nanchang, Jiangxi, China

☯ These authors contributed equally to this work.
* 2011994292@nit.edu.cn

**Data Availability Statement:** All relevant data are within the manuscript and its Supporting Information files. All files are available from the figshare database (10.6084/m9. figshare.27262470).

## Abstract

Given the multitude of risks and challenges faced by the water ecological environment during urbanization, enhancing water ecological resilience and improving the water ecological environment have emerged as crucial topics in China's economic and social development, as well as its ecological civilization construction. This study proposes a research hypothesis on the relationship between new urbanization and water ecological resilience. Employing various econometric models such as the extended STIRPAT model, dynamic panel model, panel threshold effect model, dynamic panel mediating effect model and dynamic panel difference-in-difference model, empirical tests were conducted to examine the impact of new urbanization on water ecological resilience in central China. The findings indicate that: (1) there exists a U-shaped curve relationship with a threshold effect between new urbanization and water ecological resilience; this conclusion remains valid even after conducting several robustness tests including extreme value treatment, re-measurement of independent variables, and replacement of econometric models. (2) In regions with lower levels of new urbanization, new urbanization exerts more significant stress effect on water ecological resilience through economic scale effect, population scale effect, investment pulling effect and foreign trade effect; whereas in regions with higher levels of new urbanization, new urbanization exerts more significant positive effect on water ecological resilience through factors agglomeration effect, technological progress effect, human capital effect, industrial structure effect and marketization effect. (3) Compared to non-pilot cities, the implementation of comprehensive pilot policies for new urbanization has significantly improved the water ecological resilience of cities in central China and the promotion of new urbanization of pilot cities contributes to enhancing water ecological resilience.

## Introduction

In the 14th Five-Year Plan and the Outline of Vision Goals for 2035, the Chinese government explicitly stated its commitment to enhancing the new urbanization (*NU*) strategy, advancing

**Funding:** The authors acknowledge the financial support funded by National Natural Science Foundation of China (72363022 to D.K.), Jiangxi Provincial Natural Science of Foundation (20232ACB203024 to D.K.) and Social Science Foundation of Jiangxi Province (22GL56D to D.K.). The funders had no role in study design, data collection and analysis, decision to publish, or preparation of the manuscript.

**Competing interests:** The authors have declared that no competing interests exist.

ecological civilization construction, actively promoting green development, and fostering harmonious coexistence between humanity and nature. The report from the 20th National Congress of the Communist Party of China further emphasized that coordinating water resource management, water environment preservation, and water ecology conservation is crucial for promoting such harmonious coexistence. Therefore, it is evident that integrating urbanization with water ecological environment development plays a pivotal role in achieving this goal. As urbanization progresses continuously, resources required for urbanization are partly derived from the water ecological environment, the water ecological environment faces numerous risks and challenges [1,2]. The water ecosystem is subjected to various disturbances and impacts, some of which are caused by the water ecological disorder caused by the extensive urban development. Enhancing water ecological resilience (*WER*) can effectively mitigate these pressures and threats. Consequently, this study focuses on investigating how *NU* influences *WER*.

The central China, situated in the hinterland, plays a crucial role in the country's regional economic development and serves as a vital pillar for achieving the goals of the second century. It holds an indispensable position within this context. However, rapid urbanization has led to increasingly scarce resources and mounting pressures on water ecology. This is primarily manifested by certain provinces and cities persisting with an extensive mode of economic development that requires fundamental transformation, while also necessitating further improvement in water-saving awareness. The wastage of water resources remains unresolved, resulting in low utilization efficiency. Moreover, some regions lack sufficient implementation of stringent water resource management systems and fail to effectively strengthen supervision over conservation and protection efforts. Inadequacies persist within the existing water pollution control system, leading to ongoing water environmental contamination [3,4]. In 2021, The State Council reviewed the Guiding Opinions on Promoting High-quality Development in Central China during the New Era which explicitly emphasized embracing a new path towards green and low-carbon development. This entails the economical and intensive use of energy and resource while prioritizing ecological construction and governance to achieve sustainable growth across central China. Against this backdrop, studying the impact of *NU* on *WER* in central China becomes significantly important for enhancing such resilience levels as well as addressing prevailing issues related to water environmental pollution—ultimately facilitating coordinated development among population, economy, and ecology. Currently, the academic community mainly uses the Kaya Identity, the LMDI model, and the traditional STIRPAT model to study the influencing factors of ecological environments. However, the individual factors in the decomposition of the Kaya Identity equation need to be analyzed annually or by time periods, and are subject to the constraint of maintaining an equal relationship. The LMDI model cannot examine the elasticity of each factor, while the traditional STIRPAT model only considers three aspects of population scale, wealth level and technical level, and cannot fully describe the impact of social and economic factors on ecological environments. In contrast, the STIRPAT model allows the impact of each factor to be estimated as a parameter, enabling researchers to extend the model according to their research objectives. Therefore, in order to achieve the research objectives of this paper, we use the extended STIRPAT model to analyze the impact of *NU* on *WER* in the central China, thereby providing scientific basis for formulating policies to enhance *WER* in the central China and other similar regions of the world.

## Literature review

### Research on *WER*

Holling introduced the concept of resilience in ecology, proposing a novel notion of ecological resilience that emphasizes the stable characteristics of ecosystems in response to external

disturbances and their ability to absorb such disturbances [5]. Subsequently, scholars extended this concept to urban studies, introducing the concept of urban resilience [6–8], and further derived new concepts including economic resilience, institutional resilience, social resilience, urban ecological resilience, and rural ecological resilience [9–11]. The research on ecological resilience closely related to this paper mainly encompasses two aspects: Firstly, measuring the level of ecological resilience. Existing literature has constructed various index systems based on an understanding of ecological resilience and utilized statistical data for quantitative evaluation from multiple perspectives [12,13]. Some studies have measured urban ecological resilience through considerations of urban planning and risk resistance [14,15], while others have assessed it based on land use change or landscape patterns [16,17]. Secondly is the study of factors influencing ecological resilience. Tao et al. found that urban heterogeneity exists in the impact of topographic factors, population agglomeration, opening up, industrial structure, environmental regulation and other influencing factors on ecological resilience in the Yangtze River Delta [18]. Zhang and Ren, and Wang et al. respectively analyzed the effects of provincial environmental regulation levels and urbanization intensity of Beijing-Tianjin-Hebei city cluster on the ecological resilience of China [19,20]. Based on panel data from seven major urban agglomerations in the Yellow River Basin, Wang and Niu discovered significant impacts of economic development level, scientific and technological innovation, industrial structure, population density, and environmental regulation on urban ecological resilience [21].

However, it is evident that the aforementioned literature did not discuss the *WER*. *WER* refers to the ability of water ecosystem to defend, adapt and transform under the synergistic action of artificial ecosystem and natural ecosystem when the water ecosystem is disturbed in a certain time and space. Analyzing provincial data from 2008 to 2018, Li et al. pointed out that China's *WER* has been improved, and the density index of water network and wetland rate are representative evaluation indicators affecting China's *WER* [22]. Zhang et al. investigated how pilot policies for constructing water ecological civilization cities impact *WER* [23]. Some scholars have developed an evaluation index system for measuring *WER* based on landscape ecology patterns while assessing the Hancang River basin and Linjia Village in Shandong Province. These studies primarily focus on measuring *WER* but provide limited discussion regarding its influencing factors.

## Impact of urbanization on *WER*

Currently, research on the impact of urbanization on *WER* primarily focuses on analyzing the effects of urbanization on the ecological environment [24,25] and water resources utilization [26,27]. Specifically, regarding the impact of urbanization on the ecological environment, some studies indicate that certain countries neglect the comprehensive development of urbanization, leading to exacerbated deterioration of the ecological environment. However, most scholars argue that this impact is not a simple linear relationship but rather a non-linear one. The influence of urbanization on the ecological environment depends on factors such as a country's income level, stage of urbanization, and sample interval length [28,29].

In terms of water resources utilization under urbanization, there exists abundant academic research findings. Firstly, it involves analyzing changes in water resource demand, water use structure and water pollution during the process of urbanization. Most scholars have observed that urbanization leads to increased demand for water resources along with changes in water use structure and aggravated water pollution [30–32]. Secondly, it explores the coupling relationship between urbanization and water resources as well as its impact on water resources vulnerability or carrying capacity. Some scholars argue that there is currently no strong coupling between urbanization and water resources which has hindered their coordinated

development, and further claim that urbanization has intensified water resources vulnerabilities while reducing water resources carrying capacity [33–35]. Thirdly, empirical studies have extensively examined the impact of urbanization on water resource utilization in terms of quantity, structure, and efficiency [36–38].

In summary, previous studies have provided valuable insights into understanding the measurement and influencing factors of ecological resilience, the relationship between urbanization and the ecological environment, as well as the impact of urbanization on water resource utilization. These studies have laid a strong foundation for our research. However, it is unfortunate that no existing research has explored the impact of urbanization on *WER*. While some studies have analyzed the influence of urbanization on the ecological environment, none specifically focus on *WER*. It should be noted that water resource utilization is just one aspect of the broader water ecological environment; therefore, studying the impact of urbanization on water resource utilization alone does not equate to examining overall *WER* influenced by urbanization. Consequently, this paper aims to make several potential contributions: (1) Conducting a systematic review to understand how *NU* affects *WER* and proposing corresponding research hypotheses; (2) Attempting to build various econometric models using panel data from cities in central China to empirically investigate the impact of *NU* on *WER*; (3) Utilizing a quasi-natural experiment based on comprehensive pilot projects of *NU* to further explore empirically how *NU* influences *WER* in central China through difference-in-difference method.

## Research hypothesis

After reviewing the existing literature, it is evident that *NU* has both positive and negative impacts on *WER*. On one hand, *NU* exerts stress effect on *WER*; however, on the other hand, *NU* has a positive effect on *WER*.

Firstly, the stress effect primarily arises from the fact that *NU* hampers the improvement of *WER* through various factors such as economic scale effect, population scale effect, investment pulling effect, and foreign trade effect.

1. Economic scale effect: Economic urbanization aims to expand a country's (region's) economic scale by promoting consumption and increasing supply. However, extensive economic growth leads to excessive consumption of water resources which in turn increases demand for water ecosystems. This results in water pollution and damage to the water environment while also posing threats to water security and impeding progress towards *WER* [39,40].

2. population scale effect: Population urbanization involves a significant influx of rural populations into cities leading to a substantial increase in domestic water usage and subsequent pollution. Moreover, most individuals migrating into cities find employment in labor-intensive industries or traditional service sectors that consume more water during production processes thereby exacerbating production-related pollution. Furthermore, the urbanization-driven influx of population has inflicted significant ecological damage. The escalating demand for water resources to restore, manage, and maintain the ecological environment exacerbates water security concerns and poses greater challenges to water management, thereby compromising *WER* [41,42].

3. Investment pulling effect: Spatial urbanization drives investment in urban transportation, roads, buildings, and other infrastructure. This process involves the consumption of water resources and the production of water pollution during project construction and operation. Additionally, it leads to fragmentation and encroachment on water ecological space, thereby compromising *WER* [43,44].

4. Foreign trade effect: The demographic dividend and local market effect resulting from population urbanization have attracted significant foreign capital inflows and led to a substantial increase in export scale. However, this has resulted in the consumption and pollution of water resources in the production process of foreign companies and export products. Consequently, it has negatively impacted the water ecological environment while increasing the demand for water ecosystems and undermining *WER* [45,46].

Secondly, the positive impact of *NU* on *WER* is primarily attributed to factor agglomeration effect, technological progress effect, human capital effect, industrial structure effect and marketization effect.

1. Factor agglomeration effect: Population urbanization and economic urbanization drive the concentration of labor force, capital, and other factors in cities and towns. This leads to shared water supply, water conservation, drainage, sewage treatment, water landscape, and water security facilities among a larger number of residents, enterprises, and institutions. Consequently, economies of scale are achieved which reduce the intensity of water resource consumption and pollution. Moreover, it enhances the efficiency of water resource utilization while alleviating pressure on water ecological environment and improving *WER* [47,48].

2. Technological progress effect: Economic urbanization and spatial urbanization have lowered research and development costs as well as technology promotion expenses related to technologies such as water supply, water saving, water resources recycling, water pollution treatment, reclaimed water utilization, water ecological utilization and restoration, landscape water treatment, and water safety. It is conducive to the restoration and improvement of *WER* [49–51].

3. Human capital effect: Social urbanization facilitates greater access to education, training opportunities, and better medical and health services for the migrant population. This enhances human capital levels and optimizes its allocation. Consequently, it not only improves labor productivity, reduces water consumption intensity and water resource utilization, but also enhances water resource management efficiency and adjusts water ecological responses. Simultaneously, social urbanization exposes the migrant population to urban civilization's influence, elevating overall population quality while conserving water resources, enhancing the water ecological environment, fostering a culture of sustainable water usage, and improving *WER* [26,45].

4. Industrial structure effect: Economic urbanization drives industrial structure optimization by increasing the proportion of modern manufacturing industries and emerging service sectors with lower levels of both water consumption and pollution. This contributes to enhanced efficiency in utilizing water resources while reducing water pollution intensity and water ecological environmental pressure, and improving *WER* [36,52,53].

5. Marketization effect: Economic urbanization and spatial urbanization enhance market liquidity and competitiveness, facilitating the elimination of backward production capacity resulting from market distortion. Moreover, they improve water resource allocation efficiency by addressing misplacement caused by market distortion, thereby alleviating pressure on the water ecological environment and enhancing *WER* [54,55].

In conclusion, *NU* has both positive and negative impacts on *WER*. Generally speaking, during the early stage of *NU* when its level is low, the pressure and disturbance exerted on the water ecosystem are relatively insignificant. At this time, *WER* also provides support for the

construction of *NU* at this phase. However, as *NU* progresses, its stress effect on *WER* becomes increasingly evident [56,57]. When the *NU* is further promoted, if it does not strengthen its connotation construction and change the extensive development mode, the water ecosystem will not be able to carry it, and the *NU* will be constrained by the *WER* [58,59]. Nevertheless, if adjustments are made to adopt an intensive approach with improved quality of urbanization in terms of development mode, it can promote the enhancement of *WER* in a more positive manner. Therefore, the impact of *NU* on *WER* may exhibit non-linear characteristics depending on the level of *NU*. In the early and middle stages where levels are lower, *NU* significantly affects *WER* in a detrimental way; whereas in later stages with higher levels, *NU* supports the improvement of *WER*. Accordingly, this paper proposes the following research hypotheses:

H1: A U-shaped curve relationship between *NU* and *WER* in general, with a threshold effect, is likely to exist.

H2: At lower levels of *NU*, the stress effect on *WER* becomes more pronounced through the economic scale effect, population scale effect, investment pulling effect, and foreign trade effect.

H3: At higher levels of *NU*, the positive impact on *WER* becomes more significant through the factor agglomeration effect, technological progress effect, human capital effect, industrial structure effect, and marketization effect.

## Research design

### Research area

The central China serves as a vital link between the eastern and western China, encompassing the provinces of Shanxi, Henan, Hunan, Hubei, Jiangxi, and Anhui (Fig 1). In terms of water resources, the Yangtze River traverses through Hubei, Hunan, Jiangxi, and Anhui provinces; while the Yellow River flows through Shanxi and Henan provinces in central China with abundant water resources. In 2021, the total water resources in central China were 617.95 billion $m^3$ including 586.54 billion $m^3$ of surface water and 167.8 billion $m^3$ of groundwater reserves. From a socioeconomic perspective, the central China still holds immense potential for economic development as it covers 10.7% of the country's land area and accommodates 28.1% of its population while contributing approximately 20% to national GDP. However, rapid economic growth coupled with accelerated urbanization and industrialization alongside an increasing population has led to escalating demands for water resources in this region along with worsening water pollution issues due to inadequate protection measures by certain provinces and cities within central China which consequently undermines the WER.

**Model construction.** The IPAT model serves as a prominent classical framework within environmental economics for investigating the intricate interplay between economic development and environment. Nevertheless, its applicability is significantly constrained by its inherent assumption that all factors contribute proportionally to linear changes in environmental impacts. Consequently, researchers have proposed an alternative methodology referred to as the STIRPAT model—a stochastic framework for assessing environmental impacts, specifically designed to capture the intricate dynamics underlying environmental influences, namely

$$I = aP^b A^c T^d e \tag{1}$$

Where *a* and *e* are model coefficients and error terms respectively. *I* denotes the dependent variable of ecological environment quality, while *P*, *A* and *T* refer to population scale, wealth level and technical level as independent variables respectively. *b*, *c* and *d* are parameters or

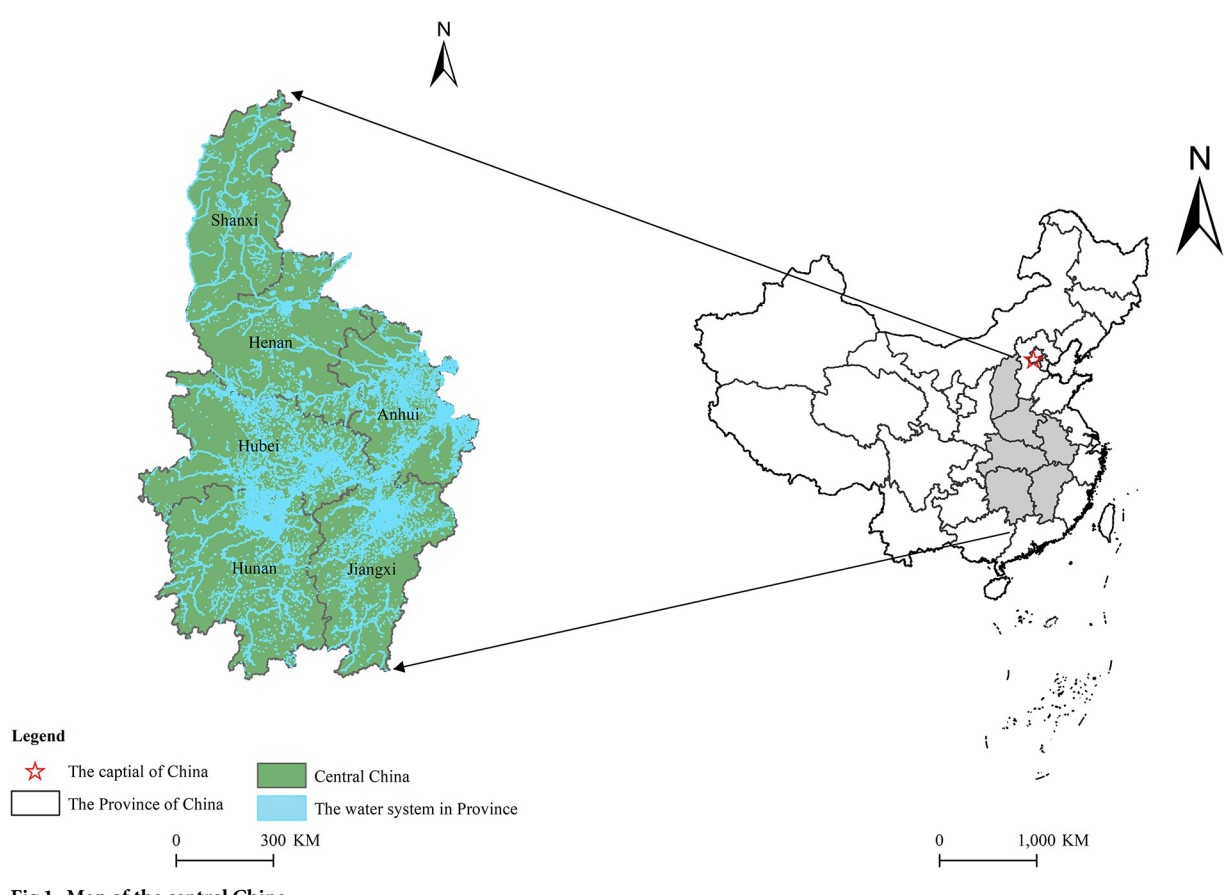

**Fig 1. Map of the central China.**

indices of the independent variables *P*, *A* and *T* correspondingly. This study aims to empirically examine the impact of *NU* on *WER*. In the STIRPAT model framework, population scale (*P*) is recognized as a factor influencing ecological environment quality (*I*). Traditional urbanization primarily focuses on increasing urban population scale and proportion; however, *NU* emphasizes connotation construction with a people-oriented approach that involves changes in both population scale and structure. Therefore, following practices of Xie et al. [60] and Jiang et al. [61] for reference purposes, population scale (*P*) is replaced by *NU* as the core independent variable in this paper. Additionally, ecological environment quality (*I*) encompasses water ecological environment quality. Enhancing *WER* can alleviate the pressure of the water ecological environment, and improve the water ecological environment. Consequently, *WER* replaces ecological environment quality (*I*) as the dependent variable in this study's modified STIRPAT model. The above STIRPAT model changes as follows:

$$WER = aNU^b A^c T^d e \tag{2}$$

Further logarithmic transformations are applied to both sides of Eq (2), yielding the following econometric model that describes the relationship between *NU* and *WER*:

$$\ln WER = a_0 + a_1 \ln NU + a_2 \ln A + a_3 \ln T + \varepsilon \tag{3}$$

Therefore, additional factors influencing the *WER* were further taken into account. Control variables such as water resource endowment (*Wb*) and climate conditions (*Qi*) were included

in the analysis. Moreover, it was acknowledged that the previous period's *WER* could have a certain impact on the current period. Consequently, an econometric model was constructed by introducing a first-order lag term of the dependent variable into the independent variable. It is as follows:

$$\ln WER_{it} = \beta_0 + \beta_1 \ln NU_{it} + \beta_2 \ln WER_{it-1} + \beta_3 \ln X_{it} + \eta_i + \delta_t + \varepsilon_{it} \tag{4}$$

Where *i* represents the city, *t* denotes the year, *X* is the control variable encompassing wealth level *A*, technical level *T*, water resource endowment *Wb*, and climate condition *Qi*. $\eta$ and $\delta$ correspond to individual fixed effects and time fixed effects respectively. To investigate the nonlinear impact of *NU* on *WER* in central China, we introduced the square term of *NU* into Eq (4). The constructed dynamic panel model is as follows:

$$\ln WER_{it} = \phi_0 + \phi_1 \ln NU_{it} + \phi_2 \ln NU_{it}^2 + \phi_3 \ln WER_{it-1} + \phi_4 \ln X_{it} + \eta_i + \delta_t + \varepsilon_{it} \tag{5}$$

In order to further investigate whether the impact of *NU* on *WER* is influenced by the stage of *NU*, i.e., if there exists a threshold effect of *NU*, this study adopts approach of Hansen and establishes a panel threshold effect model as follows [62]:

$$\ln WER_{it} = \gamma_0 + \gamma_1 \ln NU_{it} \times I(\ln NU_{it} \leq \theta) + \gamma_2 \ln NU_{it} \times I(\ln NU_{it} > \theta) + \gamma_3 X_{it} + \eta_i + \delta_t + \varepsilon_{it} \tag{6}$$

Where $\gamma_1$ and $\gamma_2$ respectively represent the parameters to be estimated for the threshold variable ln*NU* in different groups, I ($\cdot$) denotes the indicator function taking a value of 1 when the conditions in parentheses are met and 0 otherwise, and $\theta$ represents the threshold value. The threshold model can be extended to multiple threshold models according to the test results of threshold effect.

When a threshold effect occurs in the impact of *NU* on *WER*, regions with varying levels of *NU* in central China can be categorized based on the threshold value. This categorization enables analysis of the heterogeneity in the impact of different levels of *NU* on *WER* and its underlying mechanism. Specifically, we construct a dynamic panel mediation effect model as follows:

$$\ln WER_{it} = \varphi_0 + \varphi_1 \ln WER_{it-1} + \varphi_2 \ln NU_{it} + \varphi_3 \ln X_{it} + \eta_i + \delta_t + \varepsilon_{it} \tag{7}$$

$$\ln Med_{it} = \vartheta_0 + \vartheta_1 \ln Med_{it-1} + \vartheta_2 \ln NU_{it} + \vartheta_3 \ln X_{it} + \eta_i + \delta_t + \varepsilon_{it} \tag{8}$$

$$\ln WER_{it} = \lambda_0 + \lambda_1 \ln WER_{it-1} + \lambda_2 \ln NU_{it} + \lambda_3 \ln Med_{it} + \lambda_4 \ln X_{it} + \eta_i + \delta_t + \varepsilon_{it} \tag{9}$$

The impact of the independent variable *NU* on the dependent variable *WER* is tested using Eq (7). Eq (8) is employed to examine the influence of *NU* on intermediary variables (*Med*), encompassing economic scale (*Ec*), population scale (*Pe*), investment (*In*), foreign trade (*Ft*), factor agglomeration degree (*Fs*), technological progress (*Ts*), human capital (*Hu*), industrial structure (*Ms*), and marketization level (*Ma*). Eq (9) is utilized to assess the effect of the independent variable *NU* and the aforementioned intermediary variables on the dependent variable *WER*.

Utilizing a quasi-natural experiment based on comprehensive pilot projects of *NU* to further explore empirically how *NU* influences *WER* in central China through difference-in-difference method. In order to comprehensively enhance the quality of urbanization, expedite the transformation of the urbanization development mode, and promote a people-centric approach to *NU*, the National New Urbanization Plan (2014–2020) will be implemented. In 2015, the National Development and Reform Commission issued a comprehensive national pilot program for *NU* and subsequently announced three batches of national comprehensive

pilot areas for *NU*. The pilot areas including Ningbo, Dalian, Wuhan, Changsha, Chongqing are required to actively promote institutional reform and innovation while emphasizing the construction of urban ecological civilization. Their aim is to vigorously advance green and intelligent development in cities and towns. Consequently, this comprehensive pilot policy on *NU* will exert significant influence on *WER*. This policy serves as an exemplary quasi-natural experiment for studying the impact of *NU* on *WER* in central China.

The dynamic panel difference-in-difference model is constructed in this paper, with *WER* as the dependent variable and *NU*\**Zh*\**TP* as the independent variable.

$$\ln WER_{it} = \sigma_0 + \sigma_1 \ln WER_{it-1} + \sigma_2 \ln NU_{it}*Zh_i*TP_t + \sigma_3 X_{it} + \eta_i + \delta_t + \varepsilon_{it} \qquad (10)$$

Where $Zh_i$ represents the binary variable, taking a value of 1 if the city belongs to the *NU* comprehensive pilot area and 0 otherwise; $TP_t$ is a dummy variable that takes on a value of 0 prior to the implementation of the pilot policy and 1 thereafter. The remaining variables and letter meanings remain unchanged. Estimated coefficient $\sigma_2$ captures the impact of *NU* of pilot cities in the central China on *WER*. When $\sigma_2$ is significantly positive, it indicates a substantial improvement in the level of *NU* in pilot cities compared to cities that have not implemented comprehensive pilot policies for *NU* in central China. This improvement further enhances *WER* and indirectly supports research hypothesis H1.

**Variable measurement and data description.** First, to measure *NU*, this study evaluates various indicators used in existing literature and refers to studies conducted by Tang et al., Fan and Bo, Kan et al. [63–65], while considering the availability of city-level data. Indicators are selected from four dimensions of population urbanization, economic urbanization, spatial urbanization, and social urbanization to construct a comprehensive indicator system for *NU* as presented in Table 1. The range transformation method is employed to normalize the original data of each index, thereby eliminating the influence of different dimensions and orders of magnitude. Furthermore, the entropy method is utilized to determine the weights assigned to evaluation indicators for *NU* in central China, followed by applying the linear weighting method to calculate the comprehensive development index for *NU* in central China.

Secondly, the assessment of *WER* is conducted by drawing upon existing literature on urban resilience, ecological resilience, urban ecological resilience, and *WER* [23,53]. Considering the connotation and data availability of *WER*, an evaluation index system for central China's *WER* is primarily constructed based on the DPSIR model. This system encompasses five dimensions: driving forces (latent factors causing changes in *WER*), pressures (explicit factors causing changes in *WER*), states (conditions of *WER* under driving forces and pressures), influences (impacts of *WER* on society, economy, life, etc.), and responses (measures implemented by humans to enhance *WER*). It includes 29 secondary indicators such as urban population density, GDP growth rate, natural population growth rate, and sewage discharge as presented in Table 2. The measurement of *WER* in central China employs range transformation method along with entropy weight method and linear weighting method.

Third, the assessment of control variables is conducted as follows. The wealth level, technical level, water resources endowment, and climate conditions are measured using the sum of per capita disposable income of urban residents and per capita net income of rural residents after population weighting, the number of patents granted at the end of the year, the amount of per capita water resources, and precipitation respectively.

Finally, mediator variables were measured as follows: GDP, total population, total social fixed asset investment/GDP, total import and export/GDP, GDP/total labor force, non-agricultural industrial output value/GDP were used to measure economic scale, population scale, investment, foreign trade, technological progress, and industrial structure respectively. Factor

**Table 1. Index system of *NU*.**

| Target Layer | Criterion Layer | Indicator Layer |
|---|---|---|
| **New Urbanization** | Population Urbanization | Number of registered unemployed persons in urban areas (persons) |
| | | Permanent resident urbanization rate (%) |
| | | Proportion of employees in secondary industry (%) |
| | | Proportion of employees in tertiary industry (%) |
| | Economic Urbanization | GDP per capita (Yuan) |
| | | Total investment in fixed assets ($10^8$ Yuan) |
| | | Proportion of non-agricultural output value (%) |
| | | Per capita disposable income of urban residents (%) |
| | | Per capita actual utilization of foreign investment (Yuan) |
| | Spatial Urbanization | Proportion of urban construction land in urban area (%) |
| | | Park green area ($hm^2$) |
| | | Built-up area ($km^2$) |
| | | Per capita urban road area ($m^2$) |
| | | Green coverage rate of built-up area (%) |
| | | Green area ratio (%) |
| | Social Urbanization | Total retail sales of consumer goods ($10^4$ Yuan) |
| | | Number of museums (number) |
| | | Number of Internet broadband access users ($10^4$ households) |
| | | Number of hospital beds (sheets) |
| | | Urban gas penetration rate (%) |
| | | Municipal expenditure on science and technology ($10^4$ Yuan) |
| | | Education expenditure of the city ($10^4$ Yuan) |

agglomeration degree was measured by (labor factor agglomeration+ capital factor agglomeration)/2 [66], while human capital was measured by average educational level [67]. The marketization level was assessed using the marketization index compiled by the National Economic Research Institute.

The sample period for this study spans from 1997 to 2021, with the original data of GDP in central China converted using the GDP conversion index (set at 100 in 1997) to account for fluctuations in GDP. The relevant variables' original data are sourced from various publications including the China Statistical Yearbook, China Urban Statistical Yearbook, China Urban Development Report, China County (City) Socio-economic Statistical Yearbook, China Environmental Statistical Yearbook, China Urban Construction Statistical Yearbook, Water Resources Bulletin, Statistical Bulletin of National Economic and Social Development of cities in central China, and statistical yearbook of provinces in central China. In addition to these sources, data from databases such as CEIC China Economic Database and China Marketization Index Database have also been utilized. Although the marketization index time range published by the China Marketization Index Database is limited to 1997–2019; the Marketization Index is updated to 2021 based on the average annual growth rate of the data over the past years. Descriptive statistics for each variable can be found in Table 3.

## Analysis of empirical results

### Estimation results of dynamic panel model

Given that the independent variable in model (5) includes a lagged term of *WER*, it is important to note that *WER* also affect *NU*. A strong *WER* not only ensures material resources and

**Table 2. Index system of *WER*.**

| Target Layer | Criterion Layer | Indicator Layer |
|---|---|---|
| **Water Ecological Resilience** | Driving Force | Urban population density (people /km$^2$) |
| | | GDP Growth rate (%) |
| | | Natural population growth rate (%) |
| | Pressure | Sewage discharge ($10^4$ m$^3$) |
| | | Population using water ($10^4$ persons) |
| | | Water consumption of urban residents ($10^8$ m$^3$) |
| | | Total industrial water consumption ($10^8$ m$^3$) |
| | | Urban public water consumption ($10^8$ m$^3$) |
| | | Industrial wastewater discharge ($10^4$ tons) |
| | | Leakage of public water supply ($10^4$ m$^3$) |
| | State | Total water resources ($10^8$ m$^3$) |
| | | Total water supply ($10^8$ m$^3$) |
| | | Per capita water supply (m$^3$) |
| | | Water access rate (%) |
| | | Forest coverage (%) |
| | | Proportion of wetland area to national land area (%) |
| | Influence | Industrial water satisfaction (%) |
| | | Agricultural water satisfaction (%) |
| | | Domestic water satisfaction (%) |
| | | Proportion of soil erosion area (%) |
| | Response | Urban sewage treatment rate (%) |
| | | Total sewage treatment ($10^4$ tons) |
| | | Sewage treatment plant sewage treatment capacity ($10^4$ m$^3$/ day) |
| | | Industrial water reuse ($10^4$ m$^3$) |
| | | Number of wastewater treatment facilities (sets) |
| | | Drainage pipe density in built-up area (km/km$^2$) |
| | | Soil erosion control rate (%) |
| | | Proportion of water-saving irrigation area (%) |
| | | Proportion of environmental pollution control investment in GDP (%) |

spatial support for *NU* but also helps mitigate adverse effects such as crowding effect, heat island effect, rain island effect, and dry island effect during the process of urbanization. Conversely, poor *WER* can diminish environmental capacity and impose constraints on *NU* to some extent. It may even lead to negative feedback resulting in reverse urbanization phenomenon [15,68]. Consequently, endogeneity arises in model (5), necessitating the utilization of generalized method of moments (GMM) estimation techniques. Compared to difference GMM (DIF-GMM) which encounters weak instrumental problem particularly with small sample sizes; system GMM (SYS-GMM) offers smaller deviations while alleviating these issues.

**Table 3. Descriptive statistical results.**

| Variable | ln*WER* | ln*NU* | ln*A* | ln*T* | ln*Wb* | ln*Qi* | ln*Ec* | ln*Pe* | ln*In* | ln*Ft* | ln*Fs* | ln*Ts* | ln*Hu* | ln*Ms* | ln*Ma* |
|---|---|---|---|---|---|---|---|---|---|---|---|---|---|---|---|
| Mean | -0.577 | -2.407 | 9.382 | 7.371 | 7.347 | 6.932 | 6.435 | 6.094 | -0.689 | -0.895 | 1.616 | 1.142 | 2.138 | -0.271 | 1.772 |
| Median | -0.546 | -2.303 | 9.210 | 7.388 | 7.276 | 6.875 | 6.227 | 6.020 | -0.636 | -0.884 | 1.535 | 1.151 | 2.149 | -0.282 | 1.767 |
| Maximum | -0.134 | -1.138 | 10.839 | 11.367 | 8.464 | 7.526 | 9.782 | 7.219 | 0.000 | -0.109 | 2.001 | 2.708 | 2.545 | -0.016 | 2.281 |
| Minimum | -0.853 | -4.269 | 7.894 | 3.296 | 5.553 | 4.042 | 1.411 | 4.205 | -2.577 | -2.262 | 0.794 | -0.476 | 1.907 | -0.603 | 1.194 |

**Table 4. Estimation results of dynamic panel model.**

| Variable | OLS | FE | DIF-GMM | SYS-GMM |
|---|---|---|---|---|
| Constant term | 1.168 | 1.217 | 0.634[*] | 0.517[*] |
| | (0.185) | (0.256) | (0.071) | (0.073) |
| $\ln WER_{t-1}$ | 0.332 | 0.382[**] | 0.298[*] | 0.329[*] |
| | (0.207) | (0.031) | (0.083) | (0.090) |
| $\ln NU$ | -0.211[**] | -0.237[**] | -0.197[**] | -0.161[**] |
| | (0.026) | (0.043) | (0.015) | (0.034) |
| $\ln NU^2$ | 0.183[***] | 0.136[**] | 0.152[**] | 0.115[**] |
| | (0.005) | (0.014) | (0.030) | (0.026) |
| $\ln A$ | -0.154[**] | -0.135[**] | -0.096[**] | -0.103[**] |
| | (0.019) | (0.023) | (0.047) | (0.014) |
| $\ln T$ | 0.193[*] | 0.159[**] | 0.245[*] | 0.218[*] |
| | (0.087) | (0.046) | (0.072) | (0.095) |
| $\ln Wb$ | 0.149[**] | 0.160[*] | 0.129[***] | 0.086[***] |
| | (0.040) | (0.068) | (0.003) | (0.007) |
| $\ln Qi$ | 0.127[**] | 0.094[**] | 0.088[*] | 0.052[*] |
| | (0.034) | (0.029) | (0.054) | (0.081) |
| Individual fixed effect | YES | YES | YES | YES |
| Time fixed effect | YES | YES | YES | YES |
| $R^2$ | 0.926 | 0.785 | | |
| Arellano-Bond AR(1) | | | -2.542 | -2.714 |
| | | | (0.011) | (0.007) |
| Arellano-Bond AR(2) | | | 0.785 | 1.273 |
| | | | (0.392) | (0.286) |
| Sargan test | | | 9.003 | 28.158 |
| | | | (0.068) | (0.397) |

Note:

[*], [**], and [***] indicate that the variable is significant at the level of 10%, 5%, and 1%, respectively.

Therefore, this study employs SYS-GMM. The specific results are presented in Table 4 where we also provide estimates using ordinary least squares (OLS), fixed effects model (FE), and DIF-GMM respectively. It is observed that Arellano-Bond AR(1) and Arellano-Bond AR(2) statistics estimated by DIF-GMM fall within normal range; however Sargan test statistics exhibit abnormality with P-value rejecting null hypothesis indicating invalid instrumental variables used in DIF-GMM. On the other hand, both Arellano-Bond AR(1), Arellano-Bond AR (2), and Sargan test statistics estimated by SYS-GMM show no abnormalities with latter's P-value accepting null hypothesis suggesting more reliable estimations.

The findings presented in Table 4 demonstrate a significant non-linear correlation between *NU* and *WER* in central China. Specifically, the estimated coefficient of the level of *NU* is found to be negative and statistically significant at the 5% level, indicating that the *NU* has a significant negative impact on *WER* in central China. Additionally, the positive and significant coefficient of the squared term of *NU* level suggests that this relationship follows a quadratic curve pattern. In summary, our results reveal a U-shaped influence of *NU* on *WER* in central China: initially declining with increasing levels of *NU* but eventually improving beyond a certain threshold. These findings highlight that the association between *NU* and *WER* is not linear or unidirectional; instead, it exhibits a critical point threshold effect. This result is similar to the study by He et al. [69] and Zhao and Luo [70]. The former found that the coupling and

coordination degree of urbanization and ecological resilience in the Three Gorges Reservoir Area remained basically unchanged, mainly at the stage of "slight imbalance", "impending imbalance", "reluctant coordinated" or "intermediate coordination", most of which showed uncoordinated types of lagging ecological resilience or being hindered from urbanization. The latter found that, from the perspective of coupling and coordination, the grinding effect between the urbanization system and the ecological environment system in China at city level continues to be good, the number of cities with mild or lower levels of dysregulation and coordination development types is constantly decreasing. Overall, China has stepped into the stage of transformation and development. However, there is an emerging trend towards lagging ecological environments, with varying spatial distribution from south to north.

The estimation results of control variables reveal that: (1) The wealth level in central China exerts a significant negative impact on *WER*. Despite the economic development in central China, the increase in residents' income has not resulted in an improvement in *WER*; instead, it has declined. This suggests that central China is still transitioning from its high-water consumption development mode and lacks coordination between promoting water ecological environment protection and achieving high-quality development. (2) The estimated coefficient of technical level is 0.218, which is statistically significant at the 10% level, indicating a positive relationship between technical level and *WER* in central China. Enhancing the technical level within this region contributes to reducing water resource consumption intensity, improving water use efficiency, controlling water pollution, enhancing the overall quality of the water ecosystem, and fostering improved *WER*. (3) The positive influence coefficients of water resources endowment and climate conditions on *WER* in central China are statistically significant at the 1% and 10% levels, respectively. This suggests that water resources endowment and climate conditions contribute to the enhancement of *WER* in central China. However, these influences have relatively small coefficients (0.086 and 0.052), indicating that regions with abundant water resources endowment and precipitation in central China possess a larger water resources carrying capacity, resulting in strong resistance, resilience, and adaptability within the water ecosystem. Nevertheless, inadequate environmental regulations or weak implementation hinder effective responses for improving *WER* in these areas. Conversely, regions with insufficient water resources endowment and precipitation face greater pressure on their water ecological environment; however, stricter environmental regulations or more effective implementation strategies enable reasonable responses to enhance their *WER*.

The empirical results above demonstrate a U-shaped relationship between *NU* and *WER* in central China, thus confirming the first half of the H1 hypothesis. To ensure reliability, the following robustness test is conducted in this paper: (1) Extreme value processing. To mitigate the impact of extreme values on the results, this study employs a 5% bilateral winsorization method for the dependent variable and a 5% bilateral censoring method for the independent variable. The results are shown in columns (1) and (2) of Table 5. (2) Re-measure the independent variable. Firstly, we employ principal component analysis method to re-evaluate *NU*. Considering that the calculated value by principal component analysis method may be negative, we initially apply the 0–1 standardization method followed by measuring it using ln(*NU* +1). Secondly, we utilize the index system for *NU* constructed by Li and Yu to calculate its level in central China through range transformation method, entropy weight method, and linear weighting method [71]. The results are shown in columns (3) and (4) of Table 5. (3) Replace the model. The value range of *NU* and *WER* measured using range transformation method, entropy weight method, and linear weighting method, consistently falls within the interval [0,1], with truncated data attribute. The panel Tobit model is used to re-estimate, and the results are presented in column (5) of Table 5. As shown in Table 5, the estimated

**Table 5. Estimated results of robustness test.**

| Variable | Extreme value processing | | Re-measure the independent variable | | Replace the model |
|---|---|---|---|---|---|
| | (1) | (2) | (3) | (4) | (5) |
| Constant term | 0.648* | 0.579* | 0.724* | 0.589* | 1.499** |
| | (0.057) | (0.063) | (0.078) | (0.075) | (0.021) |
| $\ln WER_{t-1}$ | 0.319** | 0.304* | 0.286* | 0.297** | 0.278* |
| | (0.041) | (0.086) | (0.093) | (0.036) | (0.064) |
| $\ln NU$ | -0.156*** | -0.150** | -0.169** | -0.151** | -0.185* |
| | (0.009) | (0.037) | (0.045) | (0.028) | (0.073) |
| $\ln NU^2$ | 0.113* | 0.121** | 0.127* | 0.110** | 0.134** |
| | (0.065) | (0.018) | (0.072) | (0.034) | (0.047) |
| Control variable | YES | YES | YES | YES | YES |
| Individual fixed effect | YES | YES | YES | YES | YES |
| Time fixed effect | YES | YES | YES | YES | YES |
| Arellano-Bond AR(1) | -2.792 | -3.895 | -4.016 | -2.803 | |
| | (0.006) | (0.000) | (0.000) | (0.005) | |
| Arellano-Bond AR(2) | 1.286 | 0.587 | 0.809 | 1.264 | |
| | (0.279) | (0.541) | (0.407) | (0.291) | |
| Sargan test | 25.653 | 30.294 | 29.362 | 29.007 | |
| | (0.571) | (0.242) | (0.314) | (0.386) | |

Note:

*, **, and *** indicate that the variable is significant at the level of 10%, 5%, and 1%, respectively.

coefficient for *NU* in central China remains significantly negative while the estimated coefficient for the squared term of *NU* level continues to be significantly positive. This suggests a U-shaped relationship between *NU* and *WER* in central China, which aligns with the findings reported in Table 4, indicating that the above empirical results are robust.

## Estimation results of panel threshold effect model

The estimation results of the dynamic panel model above demonstrate a curvilinear relationship between *NU* and *WER* in central China. To further investigate potential critical points of *NU* and the heterogeneity of impacts at different critical intervals, this study employs a panel threshold effect model. After 400 times of repeated sampling, the threshold effect test results were obtained by Bootstrap self-sampling method, as shown in Table 6. These results indicate that a single threshold is significant at the 5% level (P-value = 0.049), while the double threshold is not significant (P-value = 0.637). Thus, rejecting the original hypothesis of a linear relationship between *NU* and *WER* when considering the threshold variable of *NU*. Consequently, it can be concluded that *NU* in central China exhibits a single-threshold feature on *WER*. Table 6 reveals an estimated threshold value of ln*NU* = -0.361 (*NU* = 0.697).

According to the regression results presented in Table 7, when the *NU* level falls below the threshold value of 0.697, there is a significant negative relationship between *NU* and *WER* with

**Table 6. Threshold effect test.**

| Threshold variable | Threshold number | Threshold value | F-value | P-value | Critical value (10%5%1%) | | | Number of BS |
|---|---|---|---|---|---|---|---|---|
| ln*NU* | Single threshold | -0.361 | 16.27 | 0.049 | 14.228 | 16.245 | 21.316 | 400 |
| | Double threshold | 1.613 | 6.12 | 0.637 | 20.101 | 24.359 | 29.628 | 400 |

**Table 7. Regression results of threshold effect.**

| Variable | | Coefficient estimates | P-value |
|---|---|---|---|
| 95% confidence interval | ln$NU$≤-0.361($NU$≤0.697) | -0.209[*] | 0.076 |
| [-0.368, -0.196] | ln$NU$>-0.361($NU$>0.697) | 0.156[**] | 0.043 |
| Constant term | | 3.507[**] | 0.039 |
| Control variable | | YES | |
| Individual fixed effect | | YES | |
| Time fixed effect | | YES | |
| $R^2$ | | 0.875 | |

Note

[*], [**], and [***] indicate that the variable is significant at the level of 10%, 5%, and 1%, respectively.

an estimated coefficient of -0.209 at a significance level of 10%. This suggests that during this phase, the development of NU in central China hampers improvements in WER. When the NU level exceeds the threshold value, there is a notable shift in the estimated coefficient from negative to positive for its impact on WER. The estimated coefficient becomes 0.156 and remains statistically significant at the 5% level. This finding indicates that there exists a threshold inflection point between NU and WER in central China. Once the development of NU surpasses this inflection point and enters into a high-quality stage, it significantly enhances WER. These findings highlight a distinct nonlinear "threshold" characteristic regarding how NU affects WER in central China, thereby confirming the latter part of the H1 hypothesis.

## Estimation results of dynamic panel mediation effect model

The 80 prefecture-level cities in central China were categorized based on the threshold value of the NU level. Prefecture-level cities with NU≤0.697 were classified as regions with a lower NU level (approximately 76% of the sample size), while those with NU>0.697 were classified as regions with a higher NU level (approximately 24% of the sample size). Subsequently, a dynamic panel mediation effect model was employed for estimation, and the results are presented in Tables 8–10.

Firstly, Model 1 and Model 3 in Table 8 present the estimation results of Eq (7), revealing significant variations in the impact of NU on WER across cities with different levels of NU development in central China. The estimated coefficient exhibits a significantly negative association in regions with lower levels of NU, while demonstrating a significantly positive relationship in regions with higher levels of NU. This finding further supports the notion that the connection between NU and WER follows a U-shaped pattern in central China.

Secondly, Table 9 reveals that in regions with lower levels of NU, the estimated coefficients of NU on economic scale, population scale, investment and foreign trade exhibit significant positive values with large magnitudes. This suggests that in such regions, NU plays a crucial role in fostering economic growth, expanding the population scale, driving fixed asset investment, attracting foreign investments and promoting foreign trade development. Additionally, it is observed that in regions with lower levels of NU, the estimated coefficients for factor agglomeration degree and industrial structure are positively influenced by NU; however these effects are relatively small as they pass the significance test. Conversely, the estimated coefficients for technological progress, human capital and marketization level fail to pass the significance test indicating that NU in regions with lower levels of NU there is limited impact on factor agglomeration promotion or industrial structure upgrading. Furthermore, NU fails to significantly promote technological progress or enhance human capital accumulation or

**Table 8. Impact of *NU* and intermediary variables on *WER*.**

| Variable | Regions with lower levels of *NU* | | Regions with higher levels of *NU* | |
|---|---|---|---|---|
| | Model 1 | Model 2 | Model 3 | Model 4 |
| Constant term | 1.964* | 2.239* | 3.005** | 2.412* |
| | (0.079) | (0.056) | (0.037) | (0.074) |
| ln*WER*t-1 | 0.352** | 0.201* | 0.343* | 0.228* |
| | (0.024) | (0.075) | (0.068) | (0.057) |
| ln*NU* | -0.267** | -0.183** | 0.246*** | 0.175* |
| | (0.036) | (0.040) | (0.009) | (0.081) |
| ln*Ec* | | -0.169*** | | -0.072** |
| | | (0.004) | | (0.036) |
| ln*Pe* | | -0.137* | | -0.069** |
| | | (0.058) | | (0.043) |
| ln*In* | | -0.105* | | -0.048* |
| | | (0.070) | | (0.094) |
| ln*Ft* | | -0.124** | | -0.066* |
| | | (0.036) | | (0.051) |
| ln*Fs* | | 0.048* | | 0.107* |
| | | (0.061) | | (0.065) |
| ln*Ts* | | 0.052** | | 0.139** |
| | | (0.037) | | (0.023) |
| ln*Hu* | | 0.090 | | 0.112* |
| | | (0.126) | | (0.074) |
| ln*Ms* | | 0.049* | | 0.081* |
| | | (0.078) | | (0.057) |
| ln*Ma* | | 0.083 | | 0.096* |
| | | (0.215) | | (0.063) |
| Control variable | YES | YES | YES | YES |
| Arellano-Bond AR(1) | -3.711 | -2.924 | -2.450 | -3.204 |
| | (0.000) | (0.004) | (0.008) | (0.002) |
| Arellano-Bond AR(2) | 0.563 | 1.259 | 1.416 | 0.850 |
| | (0.558) | (0.295) | (0.232) | (0.401) |
| Sargan test | 31.405 | 28.261 | 24.115 | 29.895 |
| | (0.239) | (0.388) | (0.593) | (0.336) |

improve the marketization level in regions with lower levels of *NU*. The estimation result of Eq (10) with the inclusion of intermediary variables is presented as Model 2 in Table 8. It is observed that in regions with lower levels of *NU*, the estimated coefficients for *NU*, economic scale, population scale, investment, and foreign trade on *WER* are all significantly negative. Conversely, the estimated coefficients for factor agglomeration degree, industrial structure, technological progress, human capital, and marketization level on *WER* are all positive, among these factors, only the first three estimated coefficients demonstrate statistical significance while the last two fail to do so. Based on the significance of the estimation coefficient of *NU* on the intermediary variables in Table 9, and the significant reduction of this coefficient after incorporating the intermediary variables in Table 8, it can be observed that economic scale, population scale, investment, foreign trade, factor agglomeration degree, and industrial structure act as mediators between *NU* and *WER* in regions with lower levels of *NU*. Technological progress, human capital, and marketization level do not exhibit a mediating effect between *NU* and *WER*. Therefore, in regions with lower levels of *NU* in central China, it is evident that *NU*

**Table 9. Impact of *NU* on intermediary variables (regions with lower levels of *NU*).**

| Variable | ln*Ec* | ln*Pe* | ln*In* | ln*Ft* | ln*Fs* | ln*Ts* | ln*Hu* | ln*Ms* | ln*Ma* |
|---|---|---|---|---|---|---|---|---|---|
| Constant term | 3.868** | 3.982* | 3.937* | 3.292* | 4.005** | 4.081* | 3.607* | 3.370** | 3.674* |
|  | (0.036) | (0.064) | (0.080) | (0.075) | (0.042) | (0.097) | (0.053) | (0.031) | (0.056) |
| Dependent variable lagged one period | 0.283* | 0.291* | 0.282** | 0.269* | 0.287* | 0.264** | 0.276** | 0.265* | 0.280* |
|  | (0.071) | (0.053) | (0.035) | (0.094) | (0.056) | (0.028) | (0.042) | (0.078) | (0.067) |
| ln*NU* | 0.168** | 0.187** | 0.154* | 0.115** | 0.052* | 0.039 | 0.065 | 0.049* | 0.030 |
|  | (0.029) | (0.016) | (0.079) | (0.040) | (0.083) | (0.252) | (0.190) | (0.054) | (0.161) |
| Control variable | YES | YES | YES | YES | YES | YES | YES | YES | YES |
| Arellano-Bond AR(1) | -2.756 | -3.895 | -3.163 | -2.418 | -2.886 | -3.663 | -3.021 | -2.312 | -2.759 |
|  | (0.007) | (0.000) | (0.002) | (0.009) | (0.004) | (0.000) | (0.003) | (0.010) | (0.005) |
| Arellano-Bond AR(2) | 1.335 | 0.457 | 0.882 | 1.470 | 1.307 | 0.584 | 0.909 | 1.514 | 1.333 |
|  | (0.264) | (0.561) | (0.379) | (0.217) | (0.280) | (0.528) | (0.362) | (0.216) | (0.267) |
| Sargan test | 25.112 | 31.818 | 29.267 | 23.609 | 27.668 | 30.745 | 28.447 | 22.003 | 25.786 |
|  | (0.577) | (0.216) | (0.335) | (0.598) | (0.392) | (0.241) | (0.356) | (0.629) | (0.580) |

exerts a pronounced stress impact on *WER* through effects such as economic scale effect, population scale effect, investment pulling effect, and foreign trade effect. This finding validates research hypothesis H2.

Moreover, Table 10 reveals that in regions with higher levels of *NU*, the estimated coefficients of *NU* on economic scale, population scale, investment, foreign trade, factor agglomeration degree, technological progress, human capital, industrial structure and marketization level exhibit significant positive effects. Based on the estimation results of Model 4 in Table 8, it is evident that in regions with high levels of *NU*, economic scale, population scale, investment and foreign trade do not contribute favorably to the enhancement of *WER*, all variables pass the significance test but display small estimated coefficients. Conversely, *NU*, factor agglomeration degree, technological progress, human capital, industrial structure and

**Table 10. Impact of *NU* on intermediary variables (regions with higher levels of *NU*).**

| Variable | ln*Ec* | ln*Pe* | ln*In* | ln*Ft* | ln*Fs* | ln*Ts* | ln*Hu* | ln*Ms* | ln*Ma* |
|---|---|---|---|---|---|---|---|---|---|
| Constant term | 3.048* | 3.779* | 3.724* | 3.767* | 3.659** | 3.593* | 3.528* | 3.296** | 3.700** |
|  | (0.069) | (0.090) | (0.076) | (0.061) | (0.034) | (0.055) | (0.052) | (0.030) | (0.035) |
| Dependent variable lagged one period | 0.256* | 0.271** | 0.270** | 0.275* | 0.268* | 0.274* | 0.293** | 0.259* | 0.294* |
|  | (0.085) | (0.026) | (0.033) | (0.054) | (0.067) | (0.062) | (0.041) | (0.074) | (0.052) |
| ln*NU* | 0.054** | 0.038** | 0.046* | 0.027** | 0.092** | 0.089* | 0.134* | 0.148* | 0.125* |
|  | (0.037) | (0.043) | (0.075) | (0.018) | (0.026) | (0.057) | (0.089) | (0.053) | (0.076) |
| Control variable | YES | YES | YES | YES | YES | YES | YES | YES | YES |
| Arellano-Bond AR(1) | -2.907 | -2.223 | -2.651 | -3.524 | -3.748 | -3.045 | -2.770 | -2.653 | -2.326 |
|  | (0.003) | (0.011) | (0.007) | (0.000) | (0.000) | (0.002) | (0.005) | (0.006) | (0.009) |
| Arellano-Bond AR(2) | 0.927 | 1.545 | 1.362 | 0.596 | 0.467 | 0.900 | 1.333 | 1.364 | 1.491 |
|  | (0.355) | (0.206) | (0.259) | (0.517) | (0.551) | (0.374) | (0.272) | (0.261) | (0.215) |
| Sargan test | 26.741 | 20.683 | 23.604 | 28.906 | 29.909 | 27.511 | 26.007 | 24.235 | 22.192 |
|  | (0.360) | (0.635) | (0.582) | (0.241) | (0.218) | (0.343) | (0.396) | (0.587) | (0.604) |

Note

*, **, and *** indicate that the variable is significant at the level of 10%, 5%, and 1%, respectively.

marketization level significantly promote improvements in *WER* with large estimated coefficients. Based on the significant estimation coefficient of *NU* on intermediary variables in Table 10, and the significantly reduced estimation coefficient of *NU* after incorporating intermediary variables in Table 8, it can be observed that all intermediary variables exhibit an intermediate effect between *NU* and *WER* in regions with higher levels of *NU*. However, it is noteworthy that in regions with higher levels of *NU* in central China, *NU* exerts a more pronounced positive impact on *WER* through factors agglomeration effect, technological progress effect, human capital effect, industrial structure effect, and marketization effect. This finding substantiates research hypothesis H3.

## Estimation results of dynamic panel difference-in-difference model

This study considers the cities in central China that have implemented the comprehensive pilot policy of *NU* as the treatment group, while those that have not implemented the pilot policy serve as the control group. The difference-in-difference method is employed for estimation purposes [Meeting the parallel trend hypothesis is a prerequisite for empirically evaluating policy effects using this method. To test the parallel trend hypothesis, an event analysis approach is adopted in this paper, and results indicate no significant divergence in trends between both batches of cities in either group prior to implementing the *NU* comprehensive pilot policy respectively, thus satisfying the parallel trend hypothesis]. The corresponding results are presented in Table 11. The estimated coefficients of ln*NU*\**Zh*\**TP* in models 1–4 all exhibit significant positive values, indicating a substantial improvement in urban *WER* under the *NU* comprehensive pilot policy implemented in central China compared to non-pilot cities. This implies that regardless of whether the inclusion of fixed effects and control variables, the *NU* of pilot cities in the central China contribute significantly to enhancing *WER*.

In order to assess the reliability of the aforementioned estimation results, a robustness test was conducted using the Propensity Score Match-Difference-In-Difference method (PSM-DID method) while considering other policies: (1) PSM-DID method. To mitigate the impact of sample selection bias resulting from individual heterogeneity, we employed PSM method to construct a treatment group and a control group with strong homogeneity. Specifically, we used the control variable as a covariate and performed Logit regression with the dependent variable being whether the city is a comprehensive pilot policy city for *NU* in order

**Table 11. Results of difference-in-difference estimation.**

| Variable | Model 1 | Model 2 | Model 3 | Model 4 |
|---|---|---|---|---|
| Constant term | 3.523[*] | 2.190[**] | 2.435[*] | 3.127[*] |
| | (0.084) | (0.038) | (0.096) | (0.051) |
| ln$WER_{t-1}$ | 0.261[*] | 0.307[*] | 0.284[*] | 0.315[**] |
| | (0.078) | (0.052) | (0.080) | (0.046) |
| ln*NU*\**Zh*\**TP* | 0.347[***] | 0.249[**] | 0.308[***] | 0.192[*] |
| | (0.006) | (0.025) | (0.002) | (0.083) |
| Control variable | | | YES | YES |
| Individual fixed effect | | YES | | YES |
| Time fixed effect | | YES | | YES |
| $R^2$ | 0.519 | 0.593 | 0.781 | 0.870 |
| F | 62.335[***] | 54.926[***] | 45.687[***] | 40.254[***] |

Note

[*], [**], and [***] indicate that the variable is significant at the level of 10%, 5%, and 1%, respectively.

**Table 12. Estimated results of robustness test.**

| Variable | PSM-DID | Consider other policies |
|---|---|---|
| Constant term | 3.348[**] | 2.902[*] |
|  | (0.026) | (0.074) |
| ln$WER_{t-1}$ | 0.293[*] | 0.271[**] |
|  | (0.069) | (0.035) |
| ln$NU^*Zh^*TP$ | 0.185[**] | 0.178[*] |
|  | (0.042) | (0.057) |
| Control variable | YES | YES |
| Individual fixed effect | YES | YES |
| Time fixed effect | YES | YES |
| $R^2$ | 0.847 | 0.853 |
| $F$ | 42.001[***] | 39.526[***] |

Note
[*], [**], and [***] indicate that the variable is significant at the level of 10%, 5%, and 1%, respectively.

to obtain the propensity score. The control group was then selected based on cities with closest scores. After conducting propensity score matching, we conducted balance tests to assess the effectiveness of matching. The test results indicated no significant difference between the matched treatment group and control group. Subsequently, we utilized DID method for estimation. As shown in Table 12, it can be observed that when accounting for sample selection bias, the coefficient of ln$NU^*Zh^*TP$ is significantly positive, indicating that the *NU* of pilot cities in the central China is still conducive to the improvement of *WER*. (2) Considering the potential influence of other policies, it is important to note that alongside the comprehensive pilot policy for *NU*, the government has also introduced additional initiatives such as the low carbon city pilot policy, national smart city pilot policy, and national innovative city pilot policy. The implementation of these policies may inevitably impact the implementation effect of the *NU* in the comprehensive pilot cities. Consequently, there is a possibility that the impact of *NU* on the *WER* in cities implementing these comprehensive pilot policies may have been overestimated. To address this concern, this study incorporates virtual variables representing the low-carbon city pilot policy, national smart city pilot policy, and national innovative city pilot policy into Eq (10) in order to exclude the impact of these policies on the effect of *NU* on *WER*. Upon examining the results in Table 12, it becomes evident that while the coefficient for ln$NU^*Zh^*TP$ remains significantly positive after incorporating these virtual variables, its magnitude does decrease somewhat. This suggests that although there might have been an overestimation regarding how much *NU* improves *WER* in cities implementing these comprehensive pilot policies; nevertheless, it does not alter our conclusion that *NU* of pilot cities in the central China enhances *WER*. This once again proves the robustness of the above estimates.

## Conclusions and policy recommendations

Approximately 80% of China's provinces are currently experiencing water scarcity, with significant water pollution issues exacerbating the supply-demand imbalance in the urbanization process. This has emerged as a key bottleneck hindering the advancement of *NU*. In 2022, the urbanization rate of China and developed countries differs by nearly 15 percentage points. The urbanization rate among China's registered population stands at approximately 48%, indicating a rapid advancement in the urbanization process. Consequently, enhancing the water ecological environment during this process assumes paramount importance and has emerged as a

crucial topic within China's economic development and ecological civilization construction. Embracing the concept of resilience, this study adopts a novel research perspective to address water ecological environmental challenges amidst urbanization by focusing on bolstering the resistance, resilience, and adaptability of water ecosystems. This paper initially posits a research hypothesis regarding the impact of *NU* on *WER*. Subsequently, employing panel data from cities in central China spanning from 1997 to 2021, multiple econometric models are constructed to empirically investigate the impact of *NU* on *WER* in this region. The key findings are as follows:

(1) The relationship between *NU* and *WER* in central China follows a U-shaped pattern, and this conclusion remains valid even after conducting a series of robustness tests, including extreme value treatment, re-measurement of independent variables, and replacement of econometric models. In central China, *NU* exhibits a single threshold for *WER*. Once the development of *NU* surpasses this threshold and enters the high-quality stage, significant improvements can be observed in *WER*. Therefore, it is crucial for central China to earnestly implement the strategy of *NU* while strengthening the connotation construction of urbanization from four aspects: population, economy, society, and spatial urbanization. Active promotion should be given to transforming the mode of urbanization development into one that is intensive, green, low-carbon, and intelligent. Every effort should be made to enhance the quality of urbanization by surpassing the threshold value and achieving high-quality development. This will help reduce driving forces and pressures that contribute to changes in *WER* while improving its state and enhancing response capabilities. At the same time, it is essential to comprehensively implement the strategy of ecological civilization, effectively and solidly advance the construction of water ecological civilization, integrate the management of water resources, water environment, and water ecology, timely enhance regulations on the water ecological environment, improve the efficiency of regulatory implementation, and strive to elevate the standards of water ecological protection. This approach will ensure that while pursuing *NU* development, we effectively promote the restoration and enhancement of *WER* and achieve harmonious coexistence and coordinated development between humanity and nature.

(2) In regions with lower levels of *NU* in central China, the stress effect of *NU* on *WER* is more pronounced due to the economic scale effect, population scale effect, investment pulling effect, and foreign trade effect. In regions with higher levels of *NU* in central China, *NU* has a significantly positive impact on *WER* through factor agglomeration effect, technological progress effect, human capital effect, industrial structure effect, and marketization effect. Therefore, for regions with lower levels of *NU*, it is essential to not only strengthen *NU* construction and focus on improving its quality and efficiency but also actively transform the mode of economic growth by improving comprehensive resource utilization and facilitating the development of a circular economy. Additionally, it is important to enhance population structure through refined fertility policies and population mobility strategies, optimize investment structures by increasing investments in new energy infrastructure and digital infrastructure, and adjust the foreign trade development model by leveraging digital technologies to promote digital trade. These measures aim to reduce or even reverse the stress effects caused by *NU* on *WER* through economic scale effect, population scale effect, investment pulling effect, and foreign trade effect. It is imperative to reduce the cost of factor agglomeration, establish a platform for factor agglomeration, optimize the direction of research and development investment, facilitate the commercialization of scientific and technological achievements, enhance investments in education, training, and healthcare sectors. Furthermore, it is crucial to deepen reforms in education and medical systems

while promoting the closure and transformation of industries with high water consumption and pollution levels. Additionally, efforts should be made to promote rationalization and upgrading of industrial structure through market-oriented allocation reforms for factors including water resources. Moreover, improve the operational mechanism of factor markets involving water resources. These measures can effectively amplify the positive impacts of *NU* on *WER* by leveraging factor agglomeration effect, technological progress effect, human capital effect, industrial structure effect as well as marketization effect.

(3) Compared to non-pilot cities, the implementation of comprehensive pilot policy of *NU* has significantly enhanced the urban *WER* in central China. The promotion of *WER* is facilitated by the *NU* in pilot cities. Building upon this, the experiences from central China's pilot cities are summarized and refined. Considering the heterogeneity in development levels of non-pilot cities regarding *NU*, diverse policy combinations are designed with a focus on system and mechanism innovation. Consequently, comprehensive pilot construction of *NU* is carried out in these non-pilot cities in central China through institutional improvements, enhanced standards, consolidated responsibilities, and implementation of safeguard measures. These efforts aim to enhance resistance, resilience, and adaptability of water ecosystems while elevating the overall level of *WER*.

The findings of this study can enhance the level of *WER* in the promotion of *NU* within central China. Furthermore, these results serve as a reference for other developing countries and regions akin to central China, aiming to elevate their *WER* during the advancement of *NU*. This paper improves the analysis of the impact of *NU* on *WER* and enriches the theory of *NU* and ecological resilience. The econometric model and empirical methodology employed in this study can be applied on a global scale.

## Supporting information

**S1 Data.**
(XLS)

**S1 File.**
(ZIP)

## Author Contributions

**Conceptualization:** Daxue Kan.

**Data curation:** Daxue Kan, Lianjv Lv.

**Funding acquisition:** Daxue Kan, Lianjv Lv.

**Methodology:** Lianjv Lv.

**Resources:** Daxue Kan.

**Writing – original draft:** Daxue Kan.

**Writing – review & editing:** Lianjv Lv.

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
