## [Decision Letter · Decision Letter 0]

9 Oct 2024

PONE-D-24-28533

The Impact of New Urbanization on Water Ecological Resilience: An Empirical Study from Central China

PLOS ONE

Dear Dr. Kan,

Thank you for submitting your manuscript to PLOS ONE. After careful consideration, we feel that it has merit but does not fully meet PLOS ONE’s publication criteria as it currently stands. Therefore, we invite you to submit a revised version of the manuscript that addresses the points raised during the review process.

We look forward to receiving your revised manuscript.

Kind regards,

Balaji Etikala, Ph.D.,

Academic Editor

PLOS ONE

Journal Requirements: When submitting your revision, we need you to address these additional requirements. 1. Please ensure that your manuscript meets PLOS ONE's style requirements, including those for file naming. The PLOS ONE style templates can be found at https://journals.plos.org/plosone/s/file?id=wjVg/PLOSOne_formatting_sample_main_body.pdf and https://journals.plos.org/plosone/s/file?id=ba62/PLOSOne_formatting_sample_title_authors_affiliations.pdf 2. Thank you for stating the following financial disclosure: "The authors gratefully acknowledge the financial support funded by National Natural Science Foundation of China (Grant No. 72363022) and supported by Jiangxi Provincial Natural Science of Foundation (Grant No. 20232ACB203024); This research is also supported by grants from Social Science Foundation of Jiangxi Province (Grant No. 22GL56D)."  Please state what role the funders took in the study.  If the funders had no role, please state: ""The funders had no role in study design, data collection and analysis, decision to publish, or preparation of the manuscript."" If this statement is not correct you must amend it as needed. Please include this amended Role of Funder statement in your cover letter; we will change the online submission form on your behalf. 3. Thank you for stating the following in the Acknowledgments Section of your manuscript: "The authors gratefully acknowledge the financial support funded by National Natural Science  Foundation of China (Grant No. 72363022) and supported by Jiangxi Provincial Natural Science of Foundation (Grant No. 20232ACB203024); This research is also supported by grants from Social Science Foundation of Jiangxi Province (Grant No. 22GL56D)" We note that you have provided funding information that is not currently declared in your Funding Statement. However, funding information should not appear in the Acknowledgments section or other areas of your manuscript. We will only publish funding information present in the Funding Statement section of the online submission form. Please remove any funding-related text from the manuscript and let us know how you would like to update your Funding Statement. Currently, your Funding Statement reads as follows: "The authors gratefully acknowledge the financial support funded by National Natural Science Foundation of China (Grant No. 72363022) and supported by Jiangxi Provincial Natural Science of Foundation (Grant No. 20232ACB203024); This research is also supported by grants from Social Science Foundation of Jiangxi Province (Grant No. 22GL56D)." Please include your amended statements within your cover letter; we will change the online submission form on your behalf. 4. We note that your Data Availability Statement is currently as follows: All relevant data are within the manuscript and its Supporting Information files. Please confirm at this time whether or not your submission contains all raw data required to replicate the results of your study. Authors must share the “minimal data set” for their submission. PLOS defines the minimal data set to consist of the data required to replicate all study findings reported in the article, as well as related metadata and methods (https://journals.plos.org/plosone/s/data-availability#loc-minimal-data-set-definition). For example, authors should submit the following data: - The values behind the means, standard deviations and other measures reported;- The values used to build graphs;- The points extracted from images for analysis. Authors do not need to submit their entire data set if only a portion of the data was used in the reported study. If your submission does not contain these data, please either upload them as Supporting Information files or deposit them to a stable, public repository and provide us with the relevant URLs, DOIs, or accession numbers. For a list of recommended repositories, please see https://journals.plos.org/plosone/s/recommended-repositories. If there are ethical or legal restrictions on sharing a de-identified data set, please explain them in detail (e.g., data contain potentially sensitive information, data are owned by a third-party organization, etc.) and who has imposed them (e.g., an ethics committee). Please also provide contact information for a data access committee, ethics committee, or other institutional body to which data requests may be sent. If data are owned by a third party, please indicate how others may request data access. 5. When completing the data availability statement of the submission form, you indicated that you will make your data available on acceptance. We strongly recommend all authors decide on a data sharing plan before acceptance, as the process can be lengthy and hold up publication timelines. Please note that, though access restrictions are acceptable now, your entire data will need to be made freely accessible if your manuscript is accepted for publication. This policy applies to all data except where public deposition would breach compliance with the protocol approved by your research ethics board. If you are unable to adhere to our open data policy, please kindly revise your statement to explain your reasoning and we will seek the editor's input on an exemption. Please be assured that, once you have provided your new statement, the assessment of your exemption will not hold up the peer review process. 6. Please ensure that you refer to Figure 1 in your text as, if accepted, production will need this reference to link the reader to the figure. 7. Please review your reference list to ensure that it is complete and correct. If you have cited papers that have been retracted, please include the rationale for doing so in the manuscript text, or remove these references and replace them with relevant current references. Any changes to the reference list should be mentioned in the rebuttal letter that accompanies your revised manuscript. If you need to cite a retracted article, indicate the article’s retracted status in the References list and also include a citation and full reference for the retraction notice.

**Additional Editor Comments:**

Dear authors, the script is very well written and relevant. However, following comments shall be addressed for betterment of the script.

Reviewer 1:

The manuscript presents a comprehensive analysis of the relationship between new urbanization and water ecological resilience (WER) in central China. The study is timely and relevant, given the increasing global concern for sustainable urban development and the protection of water ecosystems. The authors have employed a robust methodology, utilizing various econometric models such as the STIRPAT model, dynamic panel models, and panel threshold effect models to test their hypotheses. The use of a quasi-natural experiment based on comprehensive pilot projects for new urbanization adds a valuable layer to the analysis.

The manuscript is well-structured, with a clear abstract, introduction, literature review, methodology, results, and conclusion. The arguments are well-supported by empirical evidence, and the data analysis is thorough. The authors have also addressed potential endogeneity issues through the use of system GMM estimation, which strengthens the reliability of their findings.

In terms of minor suggestions for improvement:

1. The authors may consider providing a more detailed discussion on the policy implications of their findings, particularly how the results can inform future urban planning and water resource management strategies in central China.

2. It would be beneficial to include a comparison of the study's findings with other regional or global studies to contextualize the results within a broader framework.

Overall, the manuscript is of high quality and makes a significant contribution to the field of urbanization and environmental sustainability. I have no reservations in recommending this paper for acceptance in PLOS ONE.

Reviewer 2:

Dear editor, the script is well written and passion must be appreciated, however, I have following major and minor comments which needs to be addressed before it is considered for publication in this journal.

1. In general groundwater contains large freshwater reserves than surface water, but the authors given contradictory statement such as “In 2021, the total water resources in central China were 617.95 billion m3 including 586.54 billion m3 of surface water and 167.8billion m3 of groundwater reserves. Check once.

2. Legend in Figure 1 is not visible. Increase the font size.

3. Recheck the equations.

4. Is it typographical error “STIRPAT model---a stochastic” or it is correct expression. Check once.

5. There were several studies on STIRPAT model, emphasize this model among other models and justify selecting this model in introduction part and introduction shall end by aim of the study.

6. Is the proposed methodology can be adopted globally or it has any limitations? It shall be clearly reflected in the manuscript.

7. Comparison with other studies shall enlighten the script to further extent.

8. The authors state that “regions with low levels of NU in central China, the stress effect of NU on WER is more pronounced and regions with high levels of NU in central China, NU has a significantly positive impact on WER. IS NU stops at high level? What happens at intermediate level and what are the future implications? What is the current NU scenario in central China (high, low, or intermediate)? Shall reflect in the manuscript.

Reviewers' comments:

Reviewer's Responses to Questions

**Comments to the Author**

1. Is the manuscript technically sound, and do the data support the conclusions?

Reviewer #1: Yes

Reviewer #2: Yes

2. Has the statistical analysis been performed appropriately and rigorously? 

Reviewer #1: Yes

Reviewer #2: Yes

3. Have the authors made all data underlying the findings in their manuscript fully available?

Reviewer #1: Yes

Reviewer #2: Yes

4. Is the manuscript presented in an intelligible fashion and written in standard English?

Reviewer #1: Yes

Reviewer #2: Yes

5. Review Comments to the Author

Reviewer #1: The manuscript presents a comprehensive analysis of the relationship between new urbanization and water ecological resilience (WER) in central China. The study is timely and relevant, given the increasing global concern for sustainable urban development and the protection of water ecosystems. The authors have employed a robust methodology, utilizing various econometric models such as the STIRPAT model, dynamic panel models, and panel threshold effect models to test their hypotheses. The use of a quasi-natural experiment based on comprehensive pilot projects for new urbanization adds a valuable layer to the analysis.

The manuscript is well-structured, with a clear abstract, introduction, literature review, methodology, results, and conclusion. The arguments are well-supported by empirical evidence, and the data analysis is thorough. The authors have also addressed potential endogeneity issues through the use of system GMM estimation, which strengthens the reliability of their findings.

In terms of minor suggestions for improvement:

1. The authors may consider providing a more detailed discussion on the policy implications of their findings, particularly how the results can inform future urban planning and water resource management strategies in central China.

2. It would be beneficial to include a comparison of the study's findings with other regional or global studies to contextualize the results within a broader framework.

Overall, the manuscript is of high quality and makes a significant contribution to the field of urbanization and environmental sustainability. I have no reservations in recommending this paper for acceptance in PLOS ONE.

Reviewer #2: Dear editor, the script is well written and passion must be appreciated, however, I have following major and minor comments which needs to be addressed before it is considered for publication in this journal.

1. In general groundwater contains large freshwater reserves than surface water, but the authors given contradictory statement such as “In 2021, the total water resources in central China were 617.95 billion m3 including 586.54 billion m3 of surface water and 167.8billion m3 of groundwater reserves. Check once.

2. Legend in Figure 1 is not visible. Increase the font size.

3. Recheck the equations.

4. Is it typographical error “STIRPAT model---a stochastic” or it is correct expression. Check once.

5. There were several studies on STIRPAT model, emphasize this model among other models and justify selecting this model in introduction part and introduction shall end by aim of the study.

6. Is the proposed methodology can be adopted globally or it has any limitations? It shall be clearly reflected in the manuscript.

7. Comparison with other studies shall enlighten the script to further extent.

8. The authors state that “regions with low levels of NU in central China, the stress effect of NU on WER is more pronounced and regions with high levels of NU in central China, NU has a significantly positive impact on WER. IS NU stops at high level? What happens at intermediate level and what are the future implications? What is the current NU scenario in central China (high, low, or intermediate)? Shall reflect in the manuscript.

6. PLOS authors have the option to publish the peer review history of their article (what does this mean?). If published, this will include your full peer review and any attached files.

Reviewer #1: No

Reviewer #2: No

---

## [Author Response · Author response to Decision Letter 0]

21 Oct 2024

Author's Reply to the Review Report1

We sincerely appreciate all valuable comments and suggestions, which helped us to improve the quality of the manuscript. 

1.The authors may consider providing a more detailed discussion on the policy implications of their findings, particularly how the results can inform future urban planning and water resource management strategies in central China.

In response to the reviewer’s comment, we revised the conclusions and policy recommendations in lines 684-692 and 698-705 as follows: (1)At the same time, it is essential to comprehensively implement the strategy of ecological civilization, effectively and solidly advance the construction of water ecological civilization, integrate the management of water resources, water environment, and water ecology, timely enhance regulations on the water ecological environment, improve the efficiency of regulatory implementation, and strive to elevate the standards of water ecological protection. This approach will ensure that while pursuing NU development, we effectively promote the restoration and enhancement of WER and achieve harmonious coexistence and coordinated development between humanity and nature. (2)Therefore, for regions with lower levels of NU, it is essential to not only strengthen NU construction and focus on improving its quality and efficiency but also actively transform the mode of economic growth by improving comprehensive resource utilization and facilitating the development of a circular economy. Additionally, it is important to enhance population structure through refined fertility policies and population mobility strategies, optimize investment structures by increasing investments in new energy infrastructure and digital infrastructure, and adjust the foreign trade development model by leveraging digital technologies to promote digital trade.

2. It would be beneficial to include a comparison of the study's findings with other regional or global studies to contextualize the results within a broader framework.

In response to the reviewer’s comment, we revised analysis of empirical results in lines 455-467 and the conclusions and policy recommendations in lines 729-734 as follows: (1)This result is similar to the study by He et al. (2024) and Zhao and Luo (2024). The former found that the coupling and coordination degree of urbanization and ecological resilience in the Three Gorges Reservoir Area remained basically unchanged, mainly at the stage of “slight imbalance”, “impending imbalance”, “reluctant coordinated” or “intermediate coordination”, most of which showed uncoordinated types of lagging ecological resilience or being hindered from urbanization. The latter found that, from the perspective of coupling and coordination, the grinding effect between the urbanization system and the ecological environment system in China at city level continues to be good, the number of cities with mild or lower levels of dysregulation and coordination development types is constantly decreasing. Overall, China has stepped into the stage of transformation and development. However, there is an emerging trend towards lagging ecological environments, with varying spatial distribution from south to north. (2)The findings of this study can enhance the level of WER in the promotion of NU within central China. Furthermore, these results serve as a reference for other developing countries and regions akin to central China, aiming to elevate their WER during the advancement of NU. This paper improves the analysis of the impact of NU on WER and enriches the theory of NU and ecological resilience. The econometric model and empirical methodology employed in this study can be applied on a global scale.

Author's Reply to the Review Report2

We sincerely appreciate all valuable comments and suggestions, which helped us to improve the quality of the manuscript. 

1. In general groundwater contains large freshwater reserves than surface water, but the authors given contradictory statement such as “In 2021, the total water resources in central China were 617.95 billion m3 including 586.54 billion m3 of surface water and 167.8billion m3 of groundwater reserves. Check once.

In response to the reviewer’s comment, we once again consulted the statistical yearbook to ensure that the data was accurate. We also checked relevant data from China, as follows: In 2022, the total water resources in China were 27088.1 billion m3 including 25984.4 billion m3 of surface water and 7924.4 billion m3 of groundwater reserves. 

2. Legend in Figure 1 is not visible. Increase the font size.

In response to the reviewer’s comment, we revised Figure 1.

3. Recheck the equations.

In response to the reviewer’s comment, we rechecked the equations.

4. Is it typographical error “STIRPAT model---a stochastic” or it is correct expression. Check once.

In response to the reviewer’s comment, we revised the model construction in lines 287-289 as follows: Consequently, researchers have proposed an alternative methodology referred to as the STIRPAT model—a stochastic framework for assessing environmental impacts, specifically designed to capture the intricate dynamics underlying environmental influences.

5. There were several studies on STIRPAT model, emphasize this model among other models and justify selecting this model in introduction part and introduction shall end by aim of the study.

In response to the reviewer’s comment, we revised the introduction in lines 72-85 as follows: Currently, the academic community mainly uses the Kaya Identity, the LMDI model, and the traditional STIRPAT model to study the influencing factors of ecological environments. However, the individual factors in the decomposition of the Kaya Identity equation need to be analyzed annually or by time periods, and are subject to the constraint of maintaining an equal relationship. The LMDI model cannot examine the elasticity of each factor, while the traditional STIRPAT model only considers three aspects of population scale, wealth level and technical level, and cannot fully describe the impact of social and economic factors on ecological environments. In contrast, the STIRPAT model allows the impact of each factor to be estimated as a parameter, enabling researchers to extend the model according to their research objectives. Therefore, in order to achieve the research objectives of this paper, we use the extended STIRPAT model to analyze the impact of NU on WER in the central China, thereby providing scientific basis for formulating policies to enhance WER in the central China and other similar regions of the world.

6. Is the proposed methodology can be adopted globally or it has any limitations? It shall be clearly reflected in the manuscript.

In response to the reviewer’s comment, we revised the conclusions and policy recommendations in lines 729-734 as follows: The findings of this study can enhance the level of WER in the promotion of NU within central China. Furthermore, these results serve as a reference for other developing countries and regions akin to central China, aiming to elevate their WER during the advancement of NU. This paper improves the analysis of the impact of NU on WER and enriches the theory of NU and ecological resilience. The econometric model and empirical methodology employed in this study can be applied on a global scale.

7. Comparison with other studies shall enlighten the script to further extent.

In response to the reviewer’s comment, we revised the conclusions and policy recommendations in lines 684-692 and 698-705 as follows: (1)At the same time, it is essential to comprehensively implement the strategy of ecological civilization, effectively and solidly advance the construction of water ecological civilization, integrate the management of water resources, water environment, and water ecology, timely enhance regulations on the water ecological environment, improve the efficiency of regulatory implementation, and strive to elevate the standards of water ecological protection. This approach will ensure that while pursuing NU development, we effectively promote the restoration and enhancement of WER and achieve harmonious coexistence and coordinated development between humanity and nature. (2)Therefore, for regions with lower levels of NU, it is essential to not only strengthen NU construction and focus on improving its quality and efficiency but also actively transform the mode of economic growth by improving comprehensive resource utilization and facilitating the development of a circular economy. Additionally, it is important to enhance population structure through refined fertility policies and population mobility strategies, optimize investment structures by increasing investments in new energy infrastructure and digital infrastructure, and adjust the foreign trade development model by leveraging digital technologies to promote digital trade.

8. The authors state that “regions with low levels of NU in central China, the stress effect of NU on WER is more pronounced and regions with high levels of NU in central China, NU has a significantly positive impact on WER. IS NU stops at high level? What happens at intermediate level and what are the future implications? What is the current NU scenario in central China (high, low, or intermediate)? Shall reflect in the manuscript.

In response to the reviewer’s comment, we revised the Estimation results of dynamic panel mediation effect model in lines 544-602. The 80 prefecture-level cities in central China were categorized based on the threshold value of the NU level. Prefecture-level cities with NU≤0.697 were classified as regions with a lower NU level (approximately 76% of the sample size), while those with NU>0.697 were classified as regions with a higher NU level (approximately 24% of the sample size).

---

## [Editor Report · Decision Letter 1]

1 Nov 2024

The impact of new urbanization on water ecological resilience: An empirical study from central China

PONE-D-24-28533R1

Dear Dr. Kan,

We’re pleased to inform you that your manuscript has been judged scientifically suitable for publication and will be formally accepted for publication once it meets all outstanding technical requirements.

Kind regards,

Balaji Etikala, Ph.D.,

Academic Editor

PLOS ONE
---

## [Editor Report · Acceptance letter]

25 Nov 2024

PONE-D-24-28533R1 

PLOS ONE

Dear Dr. Kan, 

I'm pleased to inform you that your manuscript has been deemed suitable for publication in PLOS ONE. Congratulations! Your manuscript is now being handed over to our production team.

Kind regards, 

on behalf of

Dr. Balaji Etikala 

Academic Editor

PLOS ONE